# Aging-Dependent Changes in Mechanical Properties of the New Generation of Bulk-Fill Composites

**DOI:** 10.3390/ma15030902

**Published:** 2022-01-25

**Authors:** Danijela Marovic, Matej Par, Matea Macan, Nikolina Klarić, Iva Plazonić, Zrinka Tarle

**Affiliations:** 1Department of Endodontics and Restorative Dental Medicine, School of Dental Medicine, University of Zagreb, 10000 Zagreb, Croatia; mmacan@sfzg.hr (M.M.); iplazonic@sfzg.hr (I.P.); tarle@sfzg.hr (Z.T.); 2Private Dental Practice, 10000 Zagreb, Croatia; nikolinakl2511@gmail.com

**Keywords:** aging, bulk-fill, composite, ethanol, Weibull analysis, flexural strength, flexural modulus, water sorption, solubility, degree of conversion

## Abstract

This study evaluated the behavior of a new generation of bulk-fill resin composites after prolonged exposure to an aqueous environment and accelerated aging in ethanol. Six bulk-fill materials were tested (Tetric PowerFill, Filtek One Bulk Fill Restorative, Tetric EvoCeram Bulk Fill, Fill-Up!, Tetric PowerFlow, SDR Plus Bulk Fill Flowable) and compared to two conventional reference materials (Tetric EvoCeram and Tetric EvoFlow). Flexural strength, modulus, and Weibull parameters were examined at three time points: 1 day, 30 days, and 30 days followed by ethanol immersion. Degree of conversion after 30 days, water sorption, and solubility up to 90 days were also investigated. Filtek One Bulk Fill had the highest flexural strength and modulus among the tested materials, followed by Tetric PowerFill and SDR plus. Flexural strength and modulus of high-viscosity bulk-fill materials showed higher stability after accelerated aging in ethanol compared to their low-viscosity counterparts and reference materials. After 30 days, the degree of conversion was above 80% for all tested materials. Dual-cure material Fill-Up! was the best-cured material. The water sorption was highest for Fill-Up!, Filtek One Bulk Fill Restorative, and Tetric EvoFlow, while solubility was highest for Tetric EvoCeram. After aging in water and ethanol, new generation high-viscosity bulk-fill materials showed better mechanical properties than low-viscosity bulk-fill and conventional composites under extended light curing conditions.

## 1. Introduction

Proper placement of conventional composite fillings is often challenging and time-consuming. The low translucency of the material limits the thickness of the composite layer in restoration with conventional composite materials to 2 mm [1,2]. With the need to simplify the composite filling placement, bulk-fill composite materials were designed. Their advantage is the possibility of application in 4 to 5 mm layers, which allows for a shorter and simpler clinical procedure [3,4,5].

To boost a depth of cure from 2 to 4 mm without negative consequences, the composition of the composite materials had to be changed [6]. Some manufacturers have reduced the amount of glass fillers to allow deeper light penetration. In contrast, others have introduced large filler particles with a reduced surface area that minimize light reflection and scattering [7,8]. Several materials achieve high light penetration owing to the well-matched refractive indices of the filler and the organic matrix, which then changes during polymerization and acquires more opacity, desirable for the aesthetic appearance of the filling [9].

Recently, there were numerous modifications in their composition and acclaimed advantages compared to the first bulk-fill composites. The organic matrix has been thoroughly modified in specific composites to reduce polymerization stress. Very high-molecular-weight monomers with few reactive sites, such as aromatic urethane dimethacrylate (AUDMA), or monomers whose weak intramolecular bonds break at high stress—addition–fragmentation monomers—have been used. Both monomers are found in Filtek One Bulk Fill Restorative (3M, St. Paul, MN, USA; FIL) [10]. Furthermore, a clinically and experimentally very successful concept of polymerization stress reduction has been shown for a patented modified UDMA that is contained in the low-viscosity bulk-fill material SDR and its successor SDR Plus Bulk Fill Flowable (Dentsply DeTrey GmbH, Konstanz, Germany; SDR) [8].

The ambition to further shorten the duration of clinical work prompted one manufacturer of dental materials to change the composition of the existing bulk-fill composite to allow ultra-short polymerization of only 3 s with a curing unit with an extremely high light emittance of 3050 mW/cm^2^. The organic matrix of Tetric PowerFill (Ivoclar Vivadent, Schaan, Liechtenstein, PFILL) contains a β-allyl sulfone reagent, which allows the addition–fragmentation chain transfer to occur [11,12]. Instead of the uncontrolled growth of polymer chains, polymerization takes place in steps, creating short-chain polymers and more homogeneous polymerization [12,13]. Such changes in composition could reverse the negative effect of ultra-short high-intensity polymerization in older generations of composites [14]. Our previous research shows that rapid high-intensity curing did not show deleterious effects on linear shrinkage and shrinkage force [15], nor on marginal integrity [16].

In addition to the above changes in composition, bulk-fill composites also contain specially patented photoinitiators. Bis- (4-methoxybenzoyl) diethylgermanium, commercially called Ivocerin, is a Norrish I type photoinitiator based on germanium [9], which is included in PFILL and Tetric PowerFlow (Ivoclar Vivadent, PFLW). It has increased absorption of blue light in comparison to camphorquinone and provides two active radicals upon light-induced cleavage. Therefore, the germanium-based photoinitiator shortens the polymerization time, allows greater light penetration depth, and is more effective than camphorquinone [17,18]. In addition, an innovative approach to the well-known chemistry was implemented in the Fill-Up! (Coltène/Whaledent AG; Altstätten, Switzerland; COL). COL is a dual-cure bulk-fill material that, besides camphorquinone, contains benzoyl peroxide as a chemical polymerization initiator. The manufacturer states that COL can be placed in one layer of unlimited depth by using only 5 s of light activation [19].

This study examined the properties of the above-listed materials with the altered chemical composition compared to the first bulk-fill composites. In this manuscript, FIL, PFILL, PFLW, and COL are referred to as “new-generation bulk-fill composites”.

Harsh conditions in the oral cavity inevitably lead to the physical and chemical instability of restorative materials placed as a substitute for lost hard dental tissues. Over time, the mechanical properties of the composites deteriorate due to the degradation of the resin matrix, filler particles, or the silane layer at their interface [20,21]. Water penetration in dental composites can cause hydrolytic disintegration of the silane layer, followed by debonding of filler particles and even their hydrolysis [22]. The resin matrix swells and becomes plasticized upon exposure to water. The amount of water sorption depends on the available equilibrium hole-free volume and the hydrophilicity of the resin monomers [22]. The decline of the mechanical properties is considerable [23].

However, water exposure alone is not sufficient to simulate clinical conditions. Thermocycling or simple immersion in alcohol are often used to expedite aging in laboratory conditions. Organic solvents, in particular ethanol-based solutions, induce rapid degradation of mechanical properties [24]. Ethanol as a polar solvent has a similar solubility parameter to the majority of dental polymers, which facilitates its ingress into the polymer network [25]. Consequently, swelling and softening of the polymer network, as well as accelerated hydrolysis of the silane coupling agent, occur [26], additionally weakening the integrity of resin composites. In contrast, the permeability to ethanol is reduced in highly cross-linked polymer networks [25,27]. Heinze et al. found a strong correlation between a decrease in flexural strength after aging in ethanol and clinical index and a correlation between 24 h flexural strength in water and clinical wear [28]. Thus, ethanol immersion is a simple means to anticipate the long-term clinical behavior of the material [13,28].

Every manufacturer states their recommendations for light-curing times and radiant exitance. However, many studies have demonstrated improved mechanical properties after extended curing [3,29,30,31]. Such prolonged illumination with overlapping light exposures from both sides of a 2 mm thick specimen is stipulated by the ISO 4049 regulations to test flexural strength. Even though ISO recommendations are not clinically translatable, they still enable direct comparison of the materials and their behavior after prolonged exposure to an aqueous environment and ethanol-accelerated artificial aging.

Therefore, evaluation of the mechanical properties, the achieved degree of conversion, and water sorption of the new generation of bulk-fill composites is needed. The null-hypotheses were:There is no difference in the flexural strength, flexural modulus, and Weibull parameters among tested materials for the same exposure times to water and/or ethanol;There is no difference in the flexural strength, flexural modulus, and Weibull parameters for the same material and different exposure times to water and/or ethanol;There is no difference in degree of conversion among tested materials nor between two measuring depths (0 mm and 2 mm);There is no difference in water sorption and solubility among bulk-fill composites.

## 2. Materials and Methods

### 2.1. Materials

Table 1 lists the compositions of the tested materials and the recommendations on the method and duration of polymerization given by the manufacturers.

### 2.2. Testing Methods

The polymerization of tested materials was initiated using a fourth-generation polywave LED curing unit (Bluephase^®^ PowerCure, Ivoclar Vivadent) in the high power mode, with values averaging 950 mW/cm^2^. Measurement of radiant exitance was repeated three times using a hand-held radiometer (Bluephase meter, Ivoclar Vivadent, y Schaan, Liechtenstein, PFILL) before each sampling procedure, and the mean value was calculated. The emission spectrum of the curing unit had a violet peak at 404 nm and a blue light peak at 447 nm [29].

The following experiments were performed, as depicted in Figure 1:Three-point bending test according to ISO 4049 (flexural strength, flexural modulus, and Weibull analysis; 1 day, 30 days, 30 days + ethanol);Degree of conversion (30 days after light curing);Water sorption and solubility (up to 90 days).

#### 2.2.1. Specimens for Three-Point Bending Test and Degree of Conversion

Sixty specimens per material were made. For the preparation of the specimens, polished aluminum molds with a groove measuring 16 × 2 × 2 mm^3^ were used with polyethylene strips covering the top and bottom apertures. Light curing was carried out 6 times, 3 times per 20 s with radiant exitance of 950 mW/cm^2^ on each side. The consistent positioning of the light guide was ensured with a custom-made silicone mold.

The specimens were stored in plastic containers with a conical bottom (10 samples each) in 4 mL of saline solution at 37 °C in the dark.

Three measurement times were defined:1 day (n = 20);30 days (n = 20);30 days + ethanol (n = 20)—after 30 days in saline, specimens were dried and immediately immersed in 70% ethanol for 3 days.

#### 2.2.2. Three-Point Bending Test

After the designated storage time, twenty specimens were subjected to a 3-point bending test on a customized universal testing device (Ultratester, Ultradent Products Inc., South Jordan, UT, USA) at a rate of 1 mm/min. Flexural strength and modulus were calculated according to formulae described elsewhere [32].

#### 2.2.3. Degree of Conversion

An FT-Raman spectrometer (Spectrum GX, PerkinElmer, Waltham, MA, USA) was used for the collection of spectra with an NdYaG laser (wavelength 1064 nm, laser power 800 mW, resolution of 4 cm^−1^, 150 scans). Fractured specimens (n = 5) previously used in a three-point bending test in a 30-day group were dark stored in a desiccator for 3 days. Top (0 mm) and bottom surface (2 mm) was scanned. The spectra were processed, and the degree of conversion was calculated as described elsewhere [33].

#### 2.2.4. Water Sorption and Solubility

Composite specimens (n = 6) for water sorption and solubility were made using Teflon molds, dimensions 6 mm in diameter and 2 mm high. The mold was filled with unpolymerized composite material, and the top and bottom apertures were covered with polyethylene strips and pressed to remove the excess of the material. The specimens were polymerized with the same curing unit (Bluephase^®^ PowerCure, average radiant exitance 950 mW/cm^2^) for 20 s on each side.

Afterward, specimens were placed in a desiccator with freshly dried silica gel. Specimen were weighed on an analytical balance every day (NBL 254, Adam Equipment, Milton Keynes, United Kingdom) until the difference between the masses fell below 0.1 mg, i.e., until the equilibrium of the specimen mass was established. The measured value was the initial mass of the specimen (m_1_). After that, each specimen was immersed in conical bottom plastic containers with 4 mL of saline, which were dark-stored in an incubator at 37 ± 1 °C and weighed after 1, 7, and 90 days of immersion (m_2_ (t); t denotes the interval from the beginning of immersion). During the measurement, the specimens were removed from the saline solution, blot dried with absorbent paper and air-dried for 15 s. After 90 days, the specimens were again placed in a desiccator, and their mass was weighed until they reached a constant value (difference in mass between measurements of less than 0.1 mg), indicating the final mass (m_3_). The total desorption time was 26 days. Each specimen was weighed three times, and the mass of the specimen was determined by the arithmetic mean of the three measurements. Water sorption and solubility were calculated according to the equation provided by ISO 4049 [32].

### 2.3. Statistical Analysis

Since no significant deviations from the normal distribution were identified using the Shapiro–Wilk test and the inspection of normal Q–Q plots, parametric tests were used for all statistical comparisons. The assumptions of homoscedasticity and sphericity were verified using the Levene and Mauchly tests, respectively.

Required sample size was estimated in a preliminary study which showed that n = 20 is sufficient for obtaining statistical power over 0.85. A two-way mixed model ANOVA that accounted for both independent and dependent observations was used to compare flexural modulus and flexural strength for the factors “material” and “time point”. After significant effects were identified for both factors, the analysis was performed separately for individual levels of each factor. The values of flexural modulus and flexural strength among different time points within a given material were compared separately among the time points using repeated observations ANOVA followed by a Bonferroni post hoc adjustment. Weibull analysis was performed for flexural strength values to evaluate the changes in material reliability after artificial aging in water and ethanol. Weibull graphs were plotted using n = 20 data points, which were fitted to a linear function using maximum likelihood estimation [34]. The slope of thus obtained linear function represented Weibull modulus.

Degree of conversion was analyzed using a two-way ANOVA to account for the factors “material” and “depth”. As both factors exerted significant effects on the degree of conversion, the analysis was followed by one-way comparisons at individual levels of each factor. The values of the degree of conversion among materials were compared using one-way ANOVA and Tukey’s post hoc test. Comparisons of the degree of conversion at the depths of 0 mm and 2 mm within a given material were performed using independent observations *t*-test under the assumption of homoscedasticity. Water sorption and solubility values were compared among materials using one-way ANOVA and Tukey post hoc test.

The level of significance for all statistical comparisons was set to 0.05. Statistical analysis was performed using SPSS version 25 (IBM, Armonk, NY, USA).

## 3. Results

### 3.1. Mechanical Properties

Figure 2 shows that FIL had the highest flexural strength values in all time points (181 ± 35 MPa for 1 day, 182 ± 35 MPa for 30 days, and 215 ± 24 MPa for 30+eth). In the 1-day group, FIL was followed by PFILL (127 ± 12 MPa) and SDR (133 ± 18 MPa). Except for conventional reference materials (TEC and TFLW) and SDR, most materials did not show a decrease in flexural strength values after an extended period in water or water and ethanol. In contrast, some materials showed higher values after 30+eth than after 1 day, such as FIL (215 ± 24 MPa vs. 181 ± 35 MPa for 1 day), COL (130 ± 20 MPa vs. 106 ± 21 MPa for 1 day), and TBF (116 ± 14 MPa vs. 102 ± 13 MPa for 1 day). 

Figure 3 shows the highest values of the flexural modulus for FIL for all three time groups (9.0 ± 0.1 GPa for 1 day, 9.5 ± 1.5 GPa for 30 days, and 9.9 ± 1.3 GPa for 30+eth), which were statistically similar to the PFILL material after 1 day (8.4 ± 1.0 GPa), followed by other high-viscosity materials. Low-viscosity materials showed lower values in the 30+eth group, while high- and medium-viscosity materials did not show modulus degradation when 30-day and 30+eth groups were compared. The low-viscosity conventional reference material TFLW had the lowest flexural modulus (2.8 ± 0.3 GPa for 30+eth). SDR specimens stored in ethanol for 3 days after 30 days in saline had similarly low value (3.3 ± 0.3 GPa).

Weibull plots for flexural strength are shown in Figure 4. There was no systematic pattern of reliability changes as a function of aging; instead, the Weibull modulus for most of the materials was similar for all three time points (TBF, COL, PFLW, and SDR). Among these materials, the distributions of flexural strength values were either shifted toward higher values (TBF and COL) or lower values (PFLW and SDR). For PFILL and FIL, there was an increase in reliability after ethanol aging, and TFLW showed a decreased reliability with aging, while TEC showed a transient reliability decrease after 30 days, followed by similar reliability after ethanol aging, as measured after 1 day, though with the distribution of flexural strength shifted toward lower values.

### 3.2. Degree of Conversion

Figure 5 indicates that COL had the highest degree of conversion of 90%. SDR had the lowest degree of conversion of 80%. There was no difference in the degree of conversion within each material at the top (0 mm) and bottom (2 mm).

### 3.3. Water Sorption and Solubility

Figure 6 demonstrates the results of water sorption and solubility. The highest water sorption values were shown by COL, followed by FIL and TFLW, which have statistically similar results, with values above 20 μg/mm³. PFLW showed the lowest values of water absorption. TFLW, TEC, and PFILL obtained negative solubility values, while PFLW had solubility close to zero.

## 4. Discussion

This study evaluated new generation bulk-fill composites and conventional composites under extended polymerization according to ISO standards, longer than manufacturers’ recommended curing times (Table 1). Flexural strength of high-viscosity bulk-fill materials showed higher stability after extended exposure to water and accelerated aging in ethanol in comparison to their low-viscosity counterparts and the reference material. The degree of conversion in the present experimental conditions was above 80% for all tested materials, but the dual-cure material COL was the best-cured material. The water sorption was highest for COL, FIL, and TFLW, while solubility was highest for TEC. Therefore, all null hypotheses were rejected.

### 4.1. Mechanical Properties

In the present study, extended polymerization led to FIL displaying the highest flexural strength and modulus values. This was unsurprising, as FIL had the highest proportion of inorganic filler among tested materials [35]. The amount of inorganic filler is one of the most significant factors directly correlated to the flexural strength and modulus [24]. Besides, FIL was the second-best polymerized material. The strength and flexibility of the final polymer network are also of great importance for developing stable mechanical properties [36]. In our previous studies, FIL responded poorly to shortened curing times of 3 s with a high-intensity curing light. The micro- [37] and macromechanical properties [29] were jeopardized, especially in deep 4 mm layers. The recommended curing time for FIL is 20 s for 4 mm deep class I for a curing unit over 1000 mW/cm^2^. While our previous studies [29,37] used much lower radiant exitance, the present study uses six times higher values. It seems that FIL reaches its optimum properties when polymerized for longer than the manufacturer recommended time.

In this study, all high-viscosity materials had almost identical filler content by weight, about 76 wt.%. Still, FIL with 58 vol.% was the most filled material in this study and therefore expectedly showed the highest flexural strength and modulus. The other three high-viscosity materials (PFILL, TBF, and TEC) come from the same manufacturer and have a synonymous inorganic composition with about 54 vol.%. PFILL, TBF, and TEC contain prepolymerized filler particles. Such prepolymerized particles have a role in reducing the polymerization stress, better polishing, and lower water sorption but at the same time contribute little to the improvement of mechanical properties [4]. This was probably the reason for the higher values of the mechanical properties of FIL than in materials with prepolymerized particles [38]. TEC as a conventional composite displayed the lowest flexural strength of all high-viscosity materials. Randolph et al. [24] stated that materials containing prepolymerized particles have comparatively lower flexural modulus than other tested materials. In contrast, the partial replacement of filler particles with zirconium/silica filler in FIL was demonstrated as a successful strategy for improving some of the material’s mechanical properties [8].

PFILL stands out in the group of materials with high-viscosity prepolymerized fillers (PFILL, TBF, and TEC). At the 1-day measurements, the flexural modulus of PFILL was in the same statistical group with FIL, and it showed good resistance to accelerated aging by ethanol. Flexural strength of PFILL was second-best (together with SDR) after FIL after 1 day. In all high-viscosity Tetric family members tested here, the amount and the types of fillers are very similar. Hence, the apparent advantage can be assumed to originate from the resin matrix. Besides, the monomers Bis-GMA, UDMA, and Bis-EMA are common for all of them, whereas β-allyl sulfone, PBPA, and DCP are specific for PFILL [9]. DCP has a cyclic aliphatic structure [39], which probably improves mechanical properties by providing rigidity to the polymer network. Besides, the addition–fragmentation chain-transfer mechanism also influences the mechanical properties of PFILL. Gorsche et al. [11,12] demonstrated that addition–fragmentation chain-transfer reagent β-allyl sulfone homogenizes polymer networks by creating short polymer chains. This mechanism enables PFILL to be cured for only 3 s with high light energy over 3000 mW/cm^2^ with acceptable outcomes on the degree of conversion and mechanical properties [13,29,40]. However, more extended curing with moderate radiant exitance slows down polymerization kinetics [40] and further improves the crosslink density of the polymer network [13]. These conclusions are in concordance with the results of the present study.

SDR is one of the materials in which the manufacturer has decided to reduce the amount and increase the particle size [3]. The current study indicated that SDR has a high flexural strength value. The relatively small amount of filler (47 vol.%) does not explain why this material was the second-best tested material. The low flexural strength of materials containing prepolymerized fillers has most likely led to comparatively better results for SDR in the statistical analysis. Besides, the flexural modulus of SDR was lower than for other materials, as confirmed by other studies [4,36]. In the case of flexible materials, the flexural strength acquires higher values because the specimen first deforms during testing and does not crack like brittle materials, allowing for larger forces to generate before fracture. This situation probably occurred in our study. According to Fronza et al. [36], SureFil SDR flow, a precursor to the material used in this study, SDR Plus Bulk Fill Flowable, in addition to modified UDMA, also contains specific viscosity modifiers (TEGDMA, EBPDMA, and UDMA) that increase flexibility. In the current study, SDR showed the lowest degree of conversion. Consequently, this material has low or the lowest flexural modulus values in both our and previous research [36].

Mechanical properties measured after aging in water showed inconsistent results in the present study. Ferracane et al. [41] conducted a detailed investigation on mechanical properties of experimental composites with varying curing times, filler load, and silanization percentage after 2-year exposure to water. They concluded that flexural strength was the least consistent result. In that study, the material with the highest filler load, optimum silanization, and maximum curing time showed an initial decrease after 6 months, followed by an increase in flexural strength after 2 years. The flexural strength was primarily dependent on the degree of conversion and less on filler load and proper silanization. Water had little influence on flexural strength and modulus [41]. Unfortunately, studies evaluating commercial materials do not have the advantage of knowing the exact composition of the materials, which makes it challenging to draw precise conclusions.

COL, the material with the highest degree of conversion, underperformed concerning the flexural strength, showing one of the lowest values after 1 and 30 days. Low flexural strength was probably related to difficult handling in the preparation of specimens, which could have caused microscopic imperfections and voids at the specimen surface. These faults act as crack initiation sites and cause premature fracture of specimens. The reliability of COL was thus poorer than other bulk-fill materials, which is noticeable from the Weibull analysis in Figure 4.

This study evaluated the effect of aging after 30 days in water and additional accelerated aging using 3 days of exposure to 70% ethanol. Interestingly, in this study, high- and medium-viscosity materials showed an unexpected increase in flexural strength after 30 days in water and an additional 3 days in ethanol, higher than 30-day values. Such behavior was not noted for flexural strength of low-viscosity and reference materials. On the other hand, flexural modulus of high- and medium-viscosity bulk-fill materials was statistically similar in 30-day and 30+eth groups. Again, both reference and low-viscosity materials showed reduced flexural modulus after exposure to ethanol. The most probable explanation is that the higher filler ratio in high- and medium-viscosity bulk-fills diminished the amount of resin that ethanol could affect. Highly cross-linked polymer networks have less free volume available for solvent uptake and are more resistant to softening and plasticization by solvents [42]. Our previous study indirectly evaluated crosslinking density [37] of similar materials. In that study, the conventional curing (10 s, 1340 mW/cm^2^, making a radiant exposure of 13.4 J/cm^2^) resulted in higher resistance to ethanol softening of high-viscosity than low-viscosity materials, meaning that they achieved higher crosslinking density [37]. The specimens in this study received extended curing time with an extremely high radiant exposure of 114 J/cm^2^. It is reasonable to assume that high crosslinking density was achieved under such conditions even though it was not evaluated here. Besides, in highly filled composites, the tortuosity factor significantly influences the sorption kinetics of ethanol [43]. In other words, the presence of fillers diminished the diffusion coefficient for ethanol due to extended diffusion path length. The extent of the softening by ethanol is also dependent on its concentration. It is more pronounced for absolute ethanol rather than 70% ethanol diluted in water [42], as was the case in our study. We hypothesize that immersion in absolute ethanol for a longer time would provide a reduction of flexural strength and modulus even for high-viscosity materials. However, true causes of this phenomenon are still to be elucidated due to uncertainties on the precise composition of composite materials, especially as new monomers are introduced.

The behavior mentioned above is also demonstrated in the Weibull plots in Figure 4. where high-viscosity bulk-fills had a distinctive shift toward higher values for 30+eth groups (blue line) with high reliability, especially for PFILL. PFLW, on the other hand, was the material with the highest overall reliability for all groups and the narrowest distribution of the results. This can be attributed to the flowable consistency and easy handling in the specimen preparation. This fact is supported by Hayashi et al. [44], who showed that PFLW has a low tendency to form internal defects and voids.

### 4.2. Degree of Conversion

This test was conducted to ensure that all the materials received maximum polymerization on all specimen sides. Vitrification of the resin matrix after initial light curing from the top surface could have caused inhomogeneous cross-linking of polymer network and a lower degree of conversion at the bottom 2 mm of the specimens [45]. This was a concern for highly-filled composites due to high light scattering at the resin/filler interface [8]. A lower degree of conversion on the bottom could cause higher flexion of specimens before fracture at the three-point bending test [29,46]. All tested materials were polymerized according to the curing protocol recommended by ISO 4049 and showed a high degree of conversion at the sample surface and a depth of 2 mm, with no statistically significant differences between the surface and the bottom of the sample. Over 80% degree of conversion was achieved for each tested material. Therefore, the degree of conversion can be excluded as an influencing factor on flexural properties.

This result is not surprising because the degree of conversion measurements were conducted on samples prepared for the three-point bending test. According to ISO standard 4049, polymerization with mutually overlapping areas is required [32], which meant 120 s with 950 mW/cm^2^ for this study.

The highest results were achieved by COL, while SDR achieved the lowest values. COL belongs to the group of dual-cured composite materials. The surface of the filling is light-cured, and the polymerization of the entire filling is achieved by chemical initiator benzoyl peroxide [47], which is the primary reason for the high value of the degree of conversion. In addition, COL contains only 49 vol.% of fillers, reducing light scattering on the filler particles’ surface and allowing better monomer mobility [48].

### 4.3. Water Sorption and Solubility

The obtained water sorption and solubility results for all tested materials are under the maximum recommended values for composite materials [32]. The preliminary results of our water sorption study showed that 30 days are not sufficient to attain the equilibrium and to be fully saturated with water for some materials, e.g., FIL and PFILL. Therefore, we decided to prolong the exposure to 90 days.

Our study shows that TFLW, COL, and FIL had the highest amount of water absorbed over 90 days. These findings are inconsistent with COL and FIL showing the highest degree of conversion. A high degree of conversion is usually related to lower water sorption. Still, it depends on a number of variables, such as the monomer species, amount and the type of fillers, silanization, etc., [49]. The present study employed two types of specimens, bar-shaped specimens for mechanical properties and degree of conversion, and disc-shaped specimens for water sorption and solubility. The duration of water exposure and curing regimen were also different. As mentioned before, the bar-shaped specimens were excessively cured with a total radiant exposure of 114 J/cm^2^, while disc-shaped specimens received 38 J/cm^2^. Therefore, the degree of conversion of bar-shaped specimens should not be used to explain the water sorption data directly. However, the trends of a specific material’s behavior and observed differences among materials within one test are valid.

Given that the extent of water sorption depends on the filler ratio and the type of composite resin, we can conclude that the increased water absorption of COL and TFLW was caused by a reduced amount of filler, which was confirmed by most previous studies [50]. Additionally, COL contains ZnO as an antibacterial medium [47]. Therefore, ZnO at the specimen surface may attract water by forming hydrogen bonds between oxygen from ZnO and water molecules [51]. These interactions are not present in ambient temperatures, but prolonged exposure to water during 90 days at 37 °C could provide the conditions for additional water intake of COL.

The cause of the high water sorption of FIL remains unclear. The structure of two new patented monomers in their composition is not revealed; hence, their relative affinity to water is unknown. The manufacturer claims that individual silica and zirconia nano-fillers are not aggregated or agglomerated but that some silica/zirconia clusters are present. Their ratio in the composition is not disclosed [10]. Sirovica et al. recently found the microscopic heterogeneity in the degree of conversion in the polymer network surrounding individually dispersed silica fillers [52]. Unsilanized silica fillers were encapsulated with polymer of a locally lower degree of conversion, causing internal strain and stresses [52]. These events could facilitate water penetration in areas of a locally low degree of conversion. Even silanized nano-fillers tend to agglomerate. The agglomerated particles prevent the penetration of the resin, so a poorer bond is created between the filler and the resin in such a polymerized material [53]. Silane molecules, primarily if present in unbound form as found in filler particles having excessive silanization or particle agglomeration, easily hydrolyze upon exposure to water, thus creating diffusion pathways that facilitate further water sorption [54]. Previous research also considers the claim that a high proportion of fillers does not necessarily mean lower water sorption, but rather that this is determined by the properties of these fillers, i.e., whether they are unsilanized, prepolymerized, or agglomerated [39,50].

Negative solubility values, as demonstrated by TFLW, TEC, TFILL, have also been reported in other studies. The results have been explained by the formation of hydrogen bonds between the water molecules absorbed in the polymer and the polar groups of the polymer chains. Hydrogen bonds cannot be removed by dehydration and do not allow further water penetration [49,50].

## 5. Conclusions

This in vitro study showed that a new generation of high-viscosity bulk-fill resin composites have the potential to resist mechanical damage due to aging. Low-viscosity bulk-fill materials were less resistant to artificial aging in water and ethanol and should be covered with a top layer of a high-viscosity material.

The experimental conditions in this study included extended curing time, longer than manufacturers’ recommended curing duration, and 2 mm depth. Under these conditions, the trends in mechanical properties after in vitro accelerated aging were highly material-dependent. New generation bulk-fill material with addition–fragmentation chain transfer technology, Filtek One Bulk Fill, featured superior mechanical properties and a high degree of conversion, despite high water sorption. Another material with an addition–fragmentation chain transfer reagent, Tetric PowerFill, also showed significant improvement compared to its predecessor, Tetric EvoCeram Bulk Fill.

## Figures and Tables

**Figure 1 materials-15-00902-f001:**
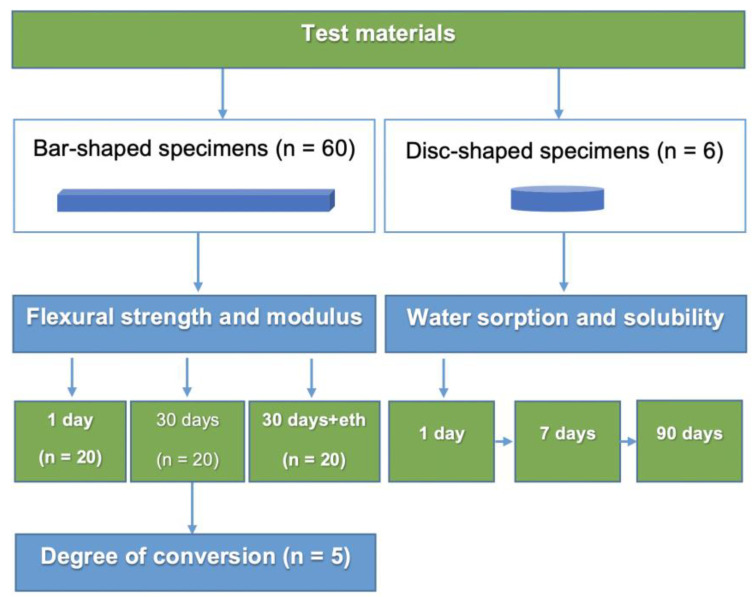
Flow chart of the experiments.

**Figure 2 materials-15-00902-f002:**
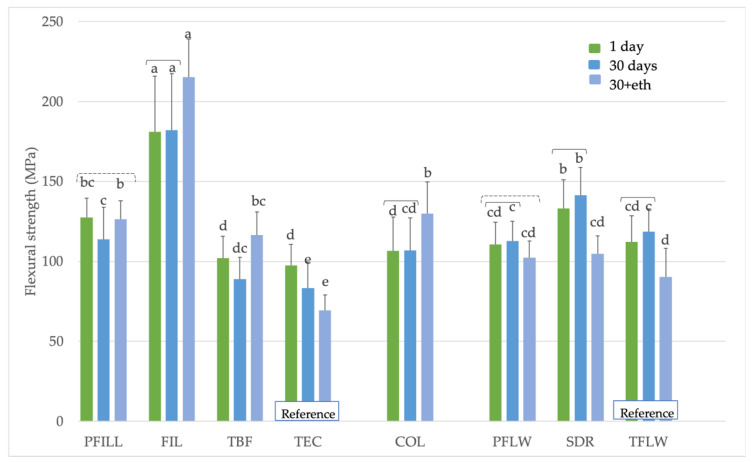
Flexural strength (MPa) of tested materials. Equal letters represent statistically homogeneous groups for a particular time point. Parentheses indicate statistically similar groups for a particular material. Dashed parentheses indicate statistically homogeneous groups within the same material, which are not similar to the column in between (in cases where the time dependence is not monotonous), *p* < 0.05.

**Figure 3 materials-15-00902-f003:**
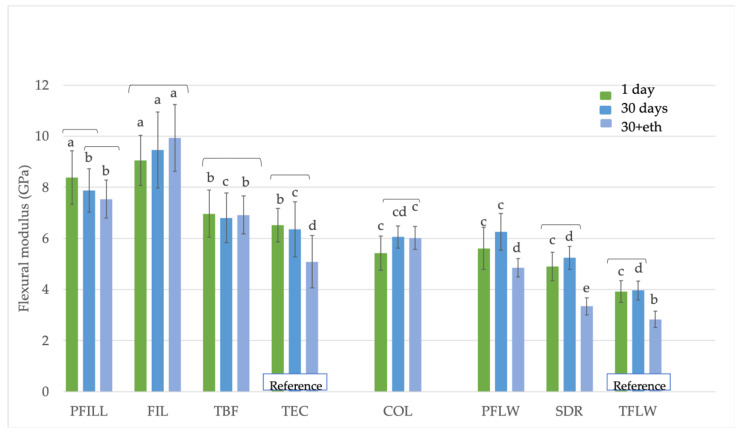
Flexural modulus (GPa) values for materials tested according to ISO 4049. Equal letters represent statistically homogeneous groups for a particular time point. Parentheses indicate statistically similar groups for each material (*p* < 0.05).

**Figure 4 materials-15-00902-f004:**
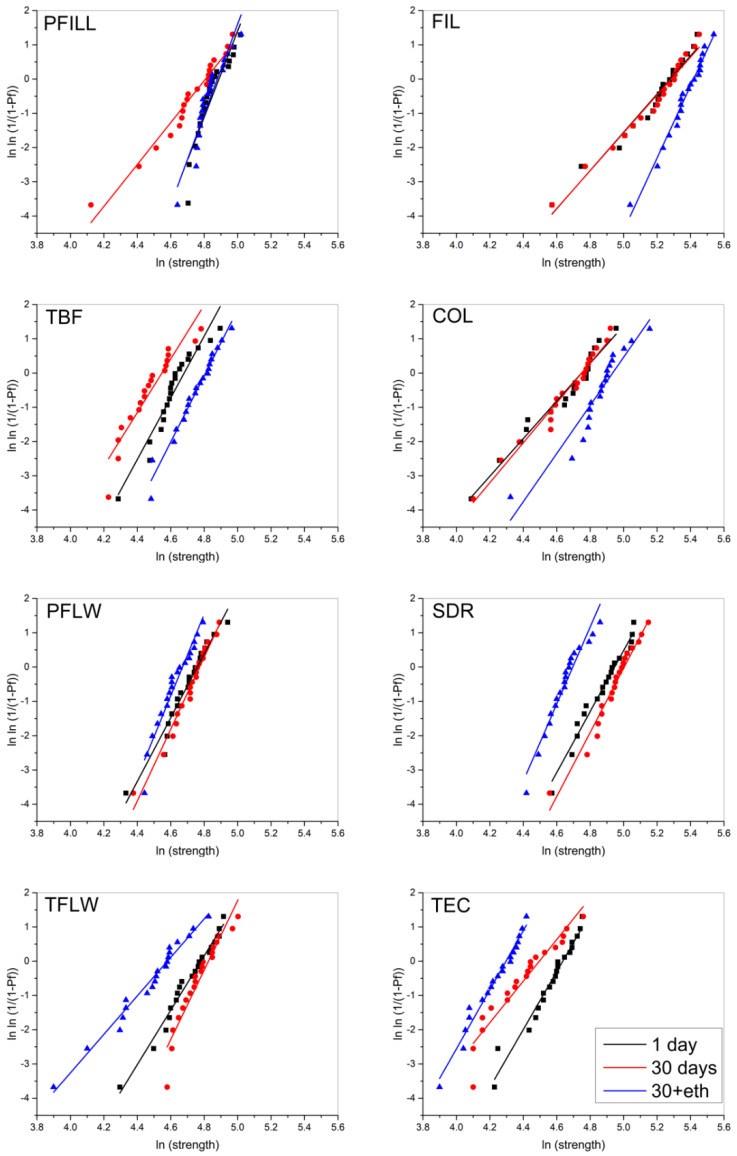
Weibull plots for the analysis of material reliability after artificial aging. Triangle, cirlcle and square represent individual time points.

**Figure 5 materials-15-00902-f005:**
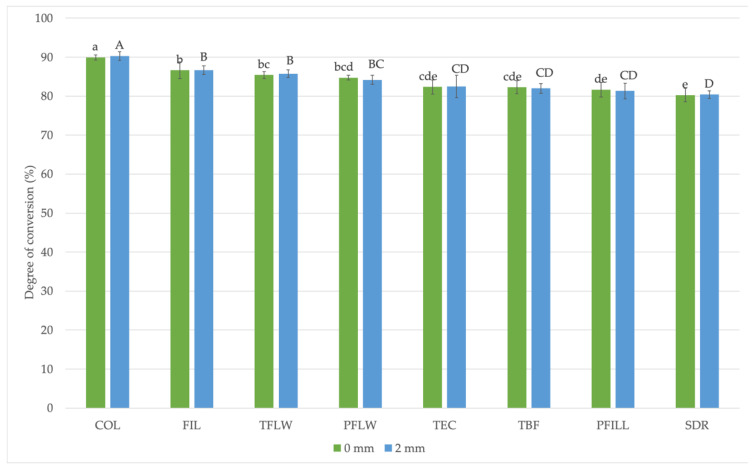
Comparison of the degree of conversion values (%) for tested materials. All pairs (0 mm vs. 2 mm) were statistically similar. Lowercase letters indicate statistically homogeneous groups at 0 mm, while uppercase letters indicate homogeneous groups at 2 mm (*p* < 0.05).

**Figure 6 materials-15-00902-f006:**
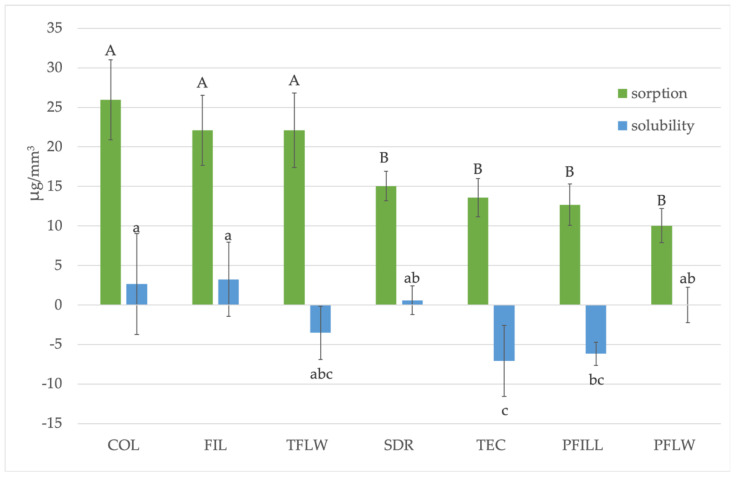
Comparison of water sorption and solubility of tested materials. Uppercase letters indicate statistically similar sorption values, while lowercase letters indicate statistically similar solubility values (*p* < 0.05).

**Table 1 materials-15-00902-t001:** Materials used in the study and the composition as provided by their respective manufacturers.

Type	Name (LOT, Abbreviation)	Organic Matrix	Filler Mass/ vol %	Manufacturer- Recommended Curing
High- viscosity	Tetric^®^ PowerFill (X56571) (PFILL)	Bis-GMA, Bis-EMA, UDMA, PBPA, DCP, β-allyl sulfone	76–77/ 53–54	3 s with 2700–3300 mW/cm^2^ 10 s with 900–1400 mW/cm^2^
	3M™ Filtek™ One Bulk Fill Restorative (NA61219) (FIL)	AUDMA, diurethane-DMA, 1,12-dodecan-DMA	~ 76.5/ ~58.5	* 20 s with 1000–2000 mW/cm^2^
	Tetric EvoCeram^®^ Bulk Fill (Y16932) (TBF)	Bis-GMA, Bis-EMA, UDMA	76–77/ 53–54	10 s with ≥1000 mW/cm^2^
	Tetric EvoCeram^®^ (Y09823) (TEC) Conventional reference	Bis-GMA, UDMA, Bis-EMA	75–76/ 53–55	10 s with ≥1000 mW/cm^2^
Medium-viscosity (dual-cure)	Fill-Up! (J57515) (COL)	methacrylates	~65/ ~49	5 s with 1600 mW/cm^2^ 10 s with 800 mW/cm^2^
Low- viscosity	Tetric^®^ PowerFlow (Y15023) (PFLW)	Bis-GMA, Bis-EMA, UDMA, DCP	68.2/ 46.4	3 s with 2700–3300 mW/cm^2^ 10 s with 900–1400 mW/cm^2^
	SDR^®^ Plus Bulk Fill Flowable (00028647) (SDR)	modified UDMA, TEGDMA, dimethacrylate, trimethacrylate resins	70.5/ 47.4	20 s with ≥550 mW/cm^2^
	Tetric EvoFlow^®^ (Y15650) (TFLW) Conventional reference	Bis-GMA, UDMA, decandiol DMA	57.5/ 30.7	10 s with ≥1000 mW/cm^2^

Bis-GMA: bisphenol A-diglycidyl dimethacrylate; Bis-EMA: ethoxylated bisphenol A dimethacrylate; DCP: tricyclodecane–dimethanol dimethacrylate; UDMA: urethane dimethacrylate; AUDMA: aromatic dimethacrylate; DMA: dimethacrylate; PBPA: propoxylated bisphenol A dimethacrylate; TEGDMA: triethylene glycol dimethacrylate. * for a LED curing unit, 4 mm layer on posterior teeth, except for class II and core build-up.

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
