# Peer review of "Aging-Dependent Changes in Mechanical Properties of the New Generation of Bulk-Fill Composites"

_materials, 2022, doi:10.3390/ma15030902_

Round 1
Reviewer 1 Report
Abstract
- The resins tested should be cited before describing the results.
- The conclusion should answer the purpose of the study and not simply try to justify the study purpose.
Introduction:
- The paragraph “In this manuscript, FIL, PFILL, PFLW, and COL are referred to as “new-generation bulk-fill composites” is lost in the introduction section.
- The use of alcohol to age must be better justify. The authors trying justify by “Exposure to water, thermocycling, or simple immersion in alcohol are often used to expedite and simulate aging in laboratory conditions. Organic solvents, in particular ethanol-based solutions induce rapid degradation of mechanical properties [24].” but this is not enough.
- If is recommended the bulk fill composites use in 4-5mm layers, why the authors tested the hypothesis: “There is no difference in degree of conversion among tested materials nor between two measuring depths (0 mm and 2 mm)”?
Materials and methods:
- The Table 1 is complete, is not necessary described the materials how was done above the Table 1 (end Page 3).
- The table 2 is confusing, the flow chart is not a Table.
- How as determined the sample size?
- Make clear the number of samples for each test.
- The ISO 4049 not preconize 90 days immersion in water, make clear this was done to age the sample.
- The evaluation time is confusing too. How long the samples were immersed in water or ethanol? 30 days? 33 days? 90 days?
- Why was choose this aging protocol? Correlate this ethanol aging with clinical behavior/relevance.
- Other confusing point about the aging: Were the samples age in ethanol only in three-point bending and degree of conversion tests?
- The statistical analysis must be revised because in study are two factors in same tests, flexural test: composite vs evalution time factors (1 day, 30days, 30+eth, see Figure 1); degree of conversion: composite vs depths…
- The discussion section is too long, consider remove some points are not relevant for your study.
Conclusion:
- Conclusion must answering the objective the study.
Reviewer 2 Report
This paper mainly introduced the research that water and ethanol ageing influences the mechanical properties of eight commercial bulk-fill dental composites which have a different level of flowability. As result, the flexural strength and modulus in high viscosity composites showed obviously higher properties before and after ageing in water and ethanol compared to other groups (medium and low viscosity composites). Nevertheless, as a kind of dental composite material, the compression properties should also be considered as an important factor and only one method (flexural test) would not clearly verify the effect of ageing on the mechanical properties (strength and modulus). I suggest adding the compression strength and modulus of the composite material in the mechanical properties part.
(1) Overall, the resulting data in f. strength/modulus and water sorption/solubility is showing a large S. deviation compared to other reference papers. It is concerned that all samples were fabricated under the same condition and were tested under a consistent procedure.
(2) It is very difficult to verify the effect of 30-days water ageing on the mechanical property changes. How about keeping 90 days or more to see the long-term effect?
(3) Why 90 days only for water sorption and solubility? Others were tested after 30-days of ageing.
(4) 3.1 Mechanical properties: It is very interesting why Fil, COL, and TBF showed higher and increased strength values after 30 days + ethanol ageing. With the short description of ‘The most probable explanation is that the higher filler ratio in high- and medium-viscosity bulk-fills diminished the amount of resin that ethanol could affect’ on page 13, it does not seem to be enough reason for the increased value of the strength after additional ageing in ethanol. Would you please suggest a more reasonable reason or kinetics?
(5) In Figure 5. There is missing data about TBS. To keep the consistency with other figures and data, the TBS group should be included.
(6) 4.2 Degree of conversion: What is the purpose of DC by comparing top and bottom after 30-days ageing? Where is the effect of aging on DC?
(7) 5. Conclusion: The conclusion is not appropriately described with the results. The summarized results and effect of ageing of the various types of composites on the mechanical and physical properties should be described in this section, but it is quite generally explained - ‘High-viscosity bulk-fill composites demonstrated better mechanical properties after water and alcohol exposure than low-viscosity materials or reference conventional composites. New ~ ~ ~.’ Also, it is not quite connected with the title ‘Aging-dependent changes in mechanical and chemical proper-ties of the new generation of bulk-fill composites.’
Reviewer 3 Report
The article is clear, well-written and presents interesting and up-to-date information about recent bulk-fill materials. The introduction is concise and presents the hypothesis in a clear way. The methods are adequate, and the experimental design is appropriate to test the hypothesis formulated. Figures and Tables are used adequately to explain the results. Discussion is adequate and compares the results with recent papers. I would recommend the article to be accepted in the present form and would like to congratulate the authors for their research.
Reviewer 4 Report
In my opinion, the work is very good, and well done, I write down some suggestions that the authors should consider to improve the work, or to discuss the process that has been done in this way.
1. In the results section, it would be good if, when describing figures 1 and 2, the mean and standard deviation could be put in brackets next to each material that they comment on.
2. If they are Bulk Fill materials that can be placed in 4-5 mm increments, why are the prepared samples 2 mm thick, which is the thickness of a conventional composite? I know that this is the thickness standardized by ISO4049, but maybe with this type of material the tests should also be carried out with the thickness that is clinically recommended for use.
3. Do you think that at a thickness of 4mm a different result could be observed in terms of the degree of conversion, when comparing the results of the top and bottom of the samples?
4. Also related to the degree of conversion, do you think that if you had measured it on samples from the water absorption study, the result would have been the same as with the bending samples? I say this because you yourselves recognise that the samples used in the bending study are over-polymerised, which may have influenced the final result of the degree of conversion. Perhaps measuring it on samples from the absorption study would have shown a little more of the reality of the material's behavior? Since in this group of samples the conventional polymerisation indications were followed.
Author Response
Dear reviewer,
We appreciate your kind words. Your comments are well placed and we have tried to correspond to them and modify the manuscript to satisfy the readers. We hope that you will be satisfied with the final outcome.
- Concern of the reviewer
In the results section, it would be good if, when describing figures 1 and 2, the mean and standard deviation could be put in brackets next to each material that they comment on.
Our response: Thank you for this comment. The mean values and standard deviations are added to the Results. Due to the comment of Reviewer 1, Figure 1 was renamed to Figure 2 and Figure 2 was renamed to Figure 3.
Revised text: “Figure 2 shows that FIL had the highest flexural strength values in all time points (180.97±34.67 MPa for 1 day, 181,9±35.36 MPa for 30 days and 215.16±23.69 MPa for 30+eth). In the 1-day group, FIL was followed by PFILL (127.48±11.96 MPa) and SDR (133.14±17.75 MPa). Except for conventional reference materials (TEC and TFLW) and SDR, most materials did not show a decrease in flexural strength values after an extended period in water or water and ethanol. In contrast, some materials showed higher values after 30+eth than after 1 day, such as FIL (215.16±23.69 MPa vs. 180.97±34.67 MPa for 1 day), COL (129.9±19.82 MPa vs. 106.46±21.27 MPa for 1 day), and TBF (116.48±14.33 MPa vs. 102.11±13.51 MPa for 1 day).”
“Figure 3 shows the highest values of the flexural modulus for FIL for all three time groups (9.05±0.98 GPa for 1 day, 9.46±1.48 GPa for 30 days and 9.93±1.31 GPa for 30+eth)., which were statistically similar to the PFILL material after 1 day (8.38±1.04 GPa), followed by other high-viscosity materials. Low-viscosity materials showed lower values in the 30+eth group, while high- and medium-viscosity materials did not show modulus degradation when 30-day and 30+eth groups were compared. The low-viscosity conventional reference material TFLW had the lowest flexural modulus (2.83±0.32 GPa for 30+eth). SDR spec-imens stored in ethanol for 3 days after 30 days in saline had a similar low value (3.34±0.33 GPa)."
- Concern of the reviewer
If they are Bulk Fill materials that can be placed in 4-5 mm increments, why are the prepared samples 2 mm thick, which is the thickness of a conventional composite? I know that this is the thickness standardized by ISO4049, but maybe with this type of material the tests should also be carried out with the thickness that is clinically recommended for use.
Our response: Thank you for this insightful comment. You have a point, and the reason why we decided not to include the degree of conversion in this manuscript was that we wanted to ensure the maximum polymerization of the specimens for the mechanical properties to avoid negative influence of low degree of conversion on the flexural strength and modulus. We made a modification in the 4.2 section of the Discussion. Besides that, our group has extensively studied the degree of conversion of bulk-fill materials in depths of 4 mm or higher. Specifically, the publication of degree of conversion of four out of eight materials tested here was recently published. Majority of these studies is cited in the submitted manuscript:
- Marovic, D.; Par, M.; Crnadak, A.; Sekelja, A.; Negovetic Mandic, V.; Gamulin, O.; Rakic, M.; Tarle, Z. Rapid 3 s Curing: What Happens in Deep Layers of New Bulk-Fill Composites? Materials (Basel) 2021, 14, doi:10.3390/ma14030515.
- Tarle, Z.; Attin, T.; Marovic, D.; Andermatt, L.; Ristic, M.; Taubock, T.T. Influence of irradiation time on subsurface degree of conversion and microhardness of high-viscosity bulk-fill resin composites. Clin Oral Investig 2015, 19, 831-840, doi:10.1007/s00784-014-1302-6.
- Marovic, D.; Taubock, T.T.; Attin, T.; Panduric, V.; Tarle, Z. Monomer conversion and shrinkage force kinetics of low-viscosity bulk-fill resin composites. Acta Odontol Scand 2015, 73, 474-480, doi:10.3109/00016357.2014.992810.
- Haugen, H.J.; Marovic, D.; Par, M.; Thieu, M.K.L.; Reseland, J.E.; Johnsen, G.F. Bulk Fill Composites Have Similar Performance to Conventional Dental Composites. Int J Mol Sci 2020, 21, doi:10.3390/ijms21145136.
- Par M, Lapas-Barisic M, Gamulin O, Panduric V, Spanovic N, Tarle Z. Long Term Degree of Conversion of two Bulk-Fill Composites. Acta Stomatol Croat. 2016 Dec;50(4):292-300. doi: 10.15644/asc50/4/2.
- Par M, Gamulin O, Marovic D, Klaric E, Tarle Z. Raman spectroscopic assessment of degree of conversion of bulk-fill resin composites--changes at 24 hours post cure. Oper Dent. 2015 May-Jun;40(3):E92-101. doi: 10.2341/14-091-L.
- Par M, Gamulin O, Marovic D, Klaric E, Tarle Z. Effect of temperature on post-cure polymerization of bulk-fill composites. J Dent. 2014 Oct;42(10):1255-60. doi: 10.1016/j.jdent.2014.08.004.
Revised text: “This test was conducted to ensure that all the materials received maximum polymerization on all specimen sides. Vitrification of the resin matrix after initial light curing from the top surface could have caused inhomogeneous cross-linking of polymer network and lower degree of conversion at the bottom 2 mm of the specimens [46]. This was a concern for highly-filled composites due to high light scattering at the resin/filler interface [8]. Lower degree if conversion on the bottom of the specimens could cause higher flexion of specimens before fracture at the three-point bending test [29,47]. All tested materials were polymerized according to the curing protocol recommended by ISO 4049 and showed a high degree of conversion at the sample surface and at a depth of 2 mm, with no statistically significant differences between the surface and the bottom of the sample. Over 80% degree of conversion was achieved for each tested material. Therefore, the degree of conversion can be excluded as an influencing factor on flexural properties.”
- Concern of the reviewer
Do you think that at a thickness of 4mm a different result could be observed in terms of the degree of conversion, when comparing the results of the top and bottom of the samples?
Our response: The 4-mm thickness would most likely result in different results. In our previous publication (Marovic et al, 2021*), we found a significantly lower degree of conversion at the bottom of the 4-mm thick specimens for similar materials as those used in this study.
* Marovic, D.; Par, M.; Crnadak, A.; Sekelja, A.; Negovetic Mandic, V.; Gamulin, O.; Rakic, M.; Tarle, Z. Rapid 3 s Curing: What Happens in Deep Layers of New Bulk-Fill Composites? Materials (Basel) 2021, 14, doi:10.3390/ma14030515.
- Concern of the reviewer
Also related to the degree of conversion, do you think that if you had measured it on samples from the water absorption study, the result would have been the same as with the bending samples? I say this because you yourselves recognise that the samples used in the bending study are over-polymerised, which may have influenced the final result of the degree of conversion. Perhaps measuring it on samples from the absorption study would have shown a little more of the reality of the material's behavior? Since in this group of samples the conventional polymerisation indications were followed.
Our response: Thank you for this interesting comment. Again, the degree of conversion would be probably lower, but not much since the water sorption specimens were also cured from top and bottom with only 2 mm thickness. Considering that Raman spectrometry requires relatively dry specimens, the water sorption specimens that were immersed in water for 90 days would have to be measured at the end of the 90-day period, which could not be correlated to the mechanical properties. On the other hand, the 90-day measurements of degree of conversion would have elucidated some issues about the water sorption and solubility and helped in Discussion.
However, considering that the purpose of the degree of conversion measurement was to serve solely as a confirmation of good polymerization of the specimens for mechanical we hope that you will be satisfied with the additional explanation given in point 1.
Round 2
Reviewer 1 Report
The paper was significantly improved.
Author Response
Dear Reviewer,
Thank you for your time and critical analysis of the manuscript that helped in it's improval.
On behalf of the authors,
Danijela Marovic
Reviewer 2 Report
Based on the title 'Aging-dependent changes in mechanical and chemical properties of the new generation of bulk-fill composites', there is no study about the effect of aging on the chemical property. Also, the correlation between water aging and mechanical property change is still unclear or not properly demonstrated with scientific/engineering methods.
Figure-4 and -5 are not directly related to the core research, they might be included in the Supplementary Information. More experimental studies and accurate analysis are required to clearly present the research purpose and to improve the quality of the manuscript to be published in Materials.
Round 3
Reviewer 2 Report
None